# Functional Identification of *Arthrinium phaeospermum* Effectors Related to *Bambusa pervariabilis* × *Dendrocalamopsis grandis* Shoot Blight

**DOI:** 10.3390/biom12091264

**Published:** 2022-09-08

**Authors:** Xinmei Fang, Peng Yan, Fengying Luo, Shan Han, Tiantian Lin, Shuying Li, Shujiang Li, Tianhui Zhu

**Affiliations:** 1College of Forestry, Sichuan Agricultural University, Chengdu 611130, China; 2Faculty of Mathematics and Natural Sciences, University of Cologne, 50674 Köln, Germany; 3National Forestry and Grassland Administration Key Laboratory of Forest Resources Conservation and Ecological Safety on the Upper Reaches of the Yangtze River, Chengdu 611130, China

**Keywords:** dual-seq, effectors, plant–pathogen interaction, gene function

## Abstract

The shoot blight of *Bambusa pervariabilis* × *Dendrocalamopsis grandis* caused by *Arthrinium phaeospermum* made bamboo die in a large area, resulting in serious ecological and economic losses. Dual RNA-seq was used to sequence and analyze the transcriptome data of *A. phaeospermum* and *B. pervariabilis* × *D. grandis* in the four periods after the pathogen infected the host and to screen the candidate effectors of the pathogen related to the infection. After the identification of the effectors by the tobacco transient expression system, the functions of these effectors were verified by gene knockout. Fifty-three differentially expressed candidate effectors were obtained by differential gene expression analysis and effector prediction. Among them, the effectors *ApCE12* and *ApCE22* can cause programmed cell death in tobacco. The disease index of *B. pervariabilis* × *D. grandis* inoculated with mutant Δ*ApCE12* and mutant Δ*ApCE22* strains were 52.5% and 47.5%, respectively, which was significantly lower than that of the wild-type strains (80%), the *ApCE12* complementary strain (77.5%), and the *ApCE22* complementary strain (75%). The tolerance of the mutant Δ*ApCE12* and mutant Δ*ApCE22* strains to H_2_O_2_ and NaCl stress was significantly lower than that of the wild-type strain and the *ApCE12* complementary and *ApCE22* complementary strains, but there was no difference in their tolerance to Congo red. Therefore, this study shows that the effectors *ApCE12* and *ApCE22* play an important role in *A. phaeospermum* virulence and response to H_2_O_2_ and NaCl stress.

## 1. Introduction

In the process of invading the host, the pathogen will be resisted by the host’s autoimmune system, including pathogen-associated molecular pattern (PAMP)-triggered immunity (PTI) and effector-triggered immunity (ETI) [1,2]. As the first plant immune barrier, PTI plays an important role in the plant’s immune response. In PTI, cell surface-localized plant pattern recognition receptors (PRRs), including receptor-like kinases (RLKs), and receptor-like proteins (RLPs), recognize PAMPs [3,4], activating immune responses, including a reactive oxygen species (ROS) burst, defense-related gene expression, and antimicrobial compound production, including chitinase secretion [5,6,7,8,9,10]. To inhibit PTI, fungal pathogens secrete effectors to escape recognition by PRRs. In turn, plants have evolved so-called resistance (R) proteins to recognize effectors and activate ETI responses [11]. In recent years, the functional identification of effectors of pathogenic fungi has become key in research on the interaction between pathogens and host plants. Although these fungal effector proteins can transfer to host cells and manipulate them to achieve infection, many effectors have no homology with known proteins in the database due to their evolutionary diversity [11,12]. Therefore, their biological functions are difficult to predict, and it is particularly important to identify the function of pathogenic effectors by related molecular techniques. Gene knockout technology is one of the most effective methods to study gene function at the gene level. Since Mishra et al. first established the transformation system of *Neurospora carssa* by gene knockout technology in 1973, genetic transformation and the identification of gene function have been completed for more than 100 filamentous fungi [13,14]. The hypersensitive response of *Nicotiana benthamiana* cells is the main feature in the response to pathogen invasion. When candidate effector proteins invade tobacco, they can cause programmed death of tobacco cells, which indicates that these candidate effector proteins may be toxin proteins or effector proteins that can be recognized by the plant’s immune system [15]. At present, the functional identification of multiple pathogen effectors has been completed by tobacco transient expression and gene knockout technology. The PgtSR1 effector of *Puccinia graminis* f.sp. tritici has been proven to be a fungal RNA silencing suppressor, which can change the abundance of small RNAs to regulate the basic defense and ETI response of plants to promote the virulence of pathogens [16]. The effector Cmu1 deletion mutant in *Ustilago maydis* greatly increased the accumulation of SA associated with a decrease in the virulence phenotype, which is considered necessary for *U**. maydis* to have virulence in maize [17].

With the rapid development of high-throughput sequencing technology, dual RNA-seq based on transcriptome sequencing technology has become an important means to study the changes in gene expression in the process of host–pathogen interaction. As a tool for studying the overall changes of expression during host–pathogen interaction, dual RNA-seq brings the feasibility for the further development of transcriptomics, which can simultaneously analyze the changes of gene expression in the pathogen and host [18]. This technology has been widely used in the study of interactions between host plants and pathogens; the powdery mildew *Erysiphe pisi* pathogen and the host plant *Medicago truncatula* [19], the *Xanthomonas oryzae* pv. Oryzicola (Xoc) pathogen and the host plant rice [20], the *Aspergillus flavus* pathogen and the host plant *Zea mays* [21], the *Xanthomonas*
*fragaria* pathogen and the host plant strawberry [22], and the *Heterobasidion annosum* s.l. pathogen and the host plant Norway spring [23] were sequenced by dual-seq. It was found that during the interaction between pathogens and hosts, the genes related to plant immune-related pathways, such as chloroplast and ROS metabolism in host plants, changed significantly, and the genes related to endoplasmic transport, toxin production, and carbohydrate metabolism in pathogens also changed significantly. Among them, in the study of the transcriptome interaction of *A**. flavus* infecting *Z**. mays*, the interaction transcripts of different stages of pathogen infection of the host were sequenced, and the different properties of *A. flavus* in the interaction transcriptome of host and pathogen in the early, middle, and late stages of maize infection were successfully captured. Then, the gene regulatory network method was used to analyze the relationships between host and pathogen genes [21,24,25]. In addition, the interaction transcriptome sequencing analysis at 12 and 29 days after the strawberry was infected by *X. fragariae* showed that the genes related to chloroplast metabolism and photosynthesis were highly expressed in the host plants in the early stage of infection, and the genes involved in host recognition and self-pathogenesis were relatively low in the pathogens in the late stage of infection. This evidence indicates the coevolution process in the interaction between pathogen and host [22]. This provides a reference and basis for the study of dual-seq in different stages of hybrid bamboo infection by *Arthrinium phaeospermum*.

*Bambusa pervariabilis* × *Dendrocalamopsis grandis* shoot blight caused by *A. phaeospermum* is a devastating disease that causes hybrid bamboo to die in a large area, resulting in serious ecological and economic losses [26]. An infection with *A. phaeospermum* has a certain impact on the physiological processes of *B. pervariabilis* × *D. grandis*, such as nucleic acid metabolism, protein metabolism, glucose metabolism [27], defense enzyme activity [28], and respiration [29]. In addition, it can also change the cell membrane permeability and destroy the tissue structure of *B. pervariabilis* × *D. grandis* [30]. At present, in addition to the physiological research, there is also some progress in molecular biology research on the disease. The whole genome expression profile of pathogenic *A. phaeospermum* provides a global view of genes and functional pathways that play a key role in the development of the disease and has resulted in valuable insights into the pathogenic molecular mechanism. On this basis, the transcriptome, proteome, and metabolome of hybrid bamboo infected by *A. phaeospermum* were sequenced, and the genes, proteins, and metabolites in response to a pathogen infection were preliminarily screened. These differentially expressed genes were mainly involved in lignin and plant protein synthesis, tetrapyrrole synthesis, redox reactions, and photosynthesis. The differentially expressed proteins were mainly concentrated in biological processes, and the four differentially expressed proteins, sucrose synthase, adenosine triphosphate citrate synthase β chain protein 1, peroxidase, and phenylalanine ammonia lyase interacted significantly with a variety of proteins. The metabolites in response to pathogen infection are mainly concentrated in metabolic pathways, such as the citric acid cycle and phenylpropane biosynthesis [31,32,33]. However, this is only an independent analysis of the changes in the expression of host plant genes, proteins, and metabolites under specific pathological conditions, and there is a lack of research on the overall changes in gene expression in the process of pathogen–host interaction. As a tool for studying the overall changes of expression in the process of host–pathogen interaction, dual-seq can simultaneously capture the transcriptome of host and pathogen, providing new biological insights into the interaction between *B. pervariabilis* × *D. grandis* and *A. phaeospermum* [18,34]. Therefore, dual RNA-seq combined with gene knockout technology was used to study susceptible plants and pathogens at the same time and analyze the function of key genes. It is of great significance to reveal the pathogenic mechanism of bacteria and plant disease resistance mechanisms, understand the regulatory network of pathogenic invasive plants, and explore the functional genes of pathogenic and host interaction

## 2. Materials and Methods

### 2.1. Strains and Plants

The micro-organisms were provided by the Laboratory of Forest Protection and Forest Pathology of Sichuan Agricultural University. The plants were two-year-old *B. pervariabilis* × *D. grandis* in the greenhouse of Sichuan Agricultural University.

### 2.2. Sample Collection and RNA Extraction

Preparation of mycelial suspension and inoculation: *A. phaeospermum* was inoculated, stored at 4 °C on PDA plates for activation, and inverted in a 25 °C incubator for 3 days. The mycelial edge was taken with an 8 mm punch, and 5 bacterial cakes were inoculated into 100 mL PDB medium containing sterile glass beads, which were shaken and cultured at 180 rpm for 5–7 d for standby. A hybrid bamboo was inoculated at a total of 20 twig bamboo nodes using sterile syringe needles, and 100 μL *A. phaeospermum* suspension was inoculated at each twig *B. pervariabilis* × *D. grandis* node, with gauze moisturizing. The plants were inoculated twice a day (9 a.m. and 9 p.m.) for 21 days.

Sampling and RNA extraction: Tender shoots of *B. pervariabilis* × *D. grandis* were frozen in liquid nitrogen at four time points before inoculation and 7, 14, and 21 days after inoculation and stored at −80 °C until use. The total RNA of pathogens and leaves was extracted by TRIzol reagent. Agarose gel electrophoresis was used to analyze the degree of degradation of RNA and whether it was contaminated. A Nanodrop microspectrophotometer (NANODROP) was used to detect the purity of the RNA (OD260/280) and quantified the RNA concentration. An Agilent 2100 was used to accurately detect the integrity of the RNA.

### 2.3. Library Construction and Sequencing

After RNA quality control was qualified, mRNA was enriched with magnetic beads with oligo (DT); fragmentation buffer was added to break it into short fragments; mRNA was used as a template; one strand of cDNA with six base random primers was synthesized, and then the buffer, dNTPs, DNA polymerase I, and RNase H were added to synthesize two strand cDNA, and the double-stranded cDNA was purified with AMPure XP beads. The purified double-stranded cDNA was repaired at the end; a tail was added and connected to the sequencing connector, and then the fragment size was selected by AMPure XP beads. Finally, PCR amplification was carried out, and the PCR products were purified with AMPure XP beads to obtain the final library. After construction of the library, Qubit 2.0 was used to preliminarily quantify and dilute the library; an Agilent 2100 was used to detect the size of the inserted fragment of the library, and RT-qPCR was used to accurately quantify the effective concentration of the library to ensure quality. Sequencing was performed by Novogene (Beijing, China) using the Illumina NovaSeq 6000 high-throughput sequencing platform.

### 2.4. Data Quality Control and Alignment with the Reference Genome

The data quality was mainly achieved by removing the low-quality, N-containing and adapter reads to obtain the clean reads required for subsequent analysis. The software HISAT (default parameters) was used to compare the scattered reads to the reference genome of the host hybrid bamboo and pathogen [35]. Among them, the reference genome of the pathogen is (*A. phaeospermum*: accession number QYRS00000000 https://www.ncbi.nlm.nih.gov/search/all/?term=QYRS00000000. Accessed on 28 June 2022). The clean hybrid bamboo transcript reads were spliced successively by Trinity v2.6 [36] software (Broad Institute, Cambridge, MA, U.S.A and Hebrew University of Jerusalem, Jerusa-lem, Israel) to obtain the reference sequence. The longest cluster sequence obtained by cross-set hierarchical clustering was analyzed, and the longest transcript Unigene spliced for *B. pervariabilis* × *D. grandis* was used as the reference sequence for subsequent analysis.

### 2.5. Gene Expression Level and Differential Expression Analyses

The gene expression level is directly proportional to the abundance of its transcripts. In this experiment, HiSeq software and a union model were used to analyze the gene expression level in each sample. According to the reads per kilobase of exon model per million mapped reads (RPKM) value of each gene, the gene expression level is calculated based on the length of the gene and the number of sequences of the gene [37]. The input data of gene differential expression analysis are the read count data obtained from gene expression level analysis. The read count is normalized; the hypothesis test probability (*p*-value) is calculated according to the model, and then multiple hypothesis test corrections are carried out to obtain the False Discovery Rate (FDR) value. When the absolute value of log2fold change is greater than 1 and the *p*-value is less than 0.05, it is considered a differential gene.

### 2.6. GO and KEGG Enrichment Analysis of Differentially Expressed Genes in Pathogens and Hosts

GeneOntology (GO) function annotation was performed according to the sequence information of different genes, including localization to biological processes, cellular components, and molecular functions. GO enrichment analysis of differentially expressed genes was carried out using the Goseq software package, and Kyoto Encyclopedia of Genes and Genomes (KEGGs) enrichment analysis of differentially expressed genes was carried out using KOBAS 2.0 software (Peking University, Beijing, China) [38,39]. The main biochemical metabolic pathway and signal transduction pathway involved in differentially expressed genes could be determined by path significant enrichment.

### 2.7. Association Analysis between Pathogens and Hosts

First, based on the different analysis results of *A. phaeospermum* (or *B. pervariabilis* × *D. grandis*), all differential genes of all samples were obtained by merging the differential genes in all comparison combinations, and the differentially expressed genes of *A. phaeospermum* (or *B. pervariabilis* × *D. grandis*) were clustered by K-means [40]. These differentially expressed genes were annotated in the pathogen–host interactions (PHIs) [41], virus factors database (VFDB, http://www.mgc.ac.cn/VFs/main.htm. Accessed on 6 September 2022) [42], and other databases, and the interaction between pathogen and host was analyzed in the search tool for the retrieval of the interaction gene (STRING, https://string-db.org/. Accessed on 1 September 2022) database [43] and host–pathogen interaction (HPIDB) [44] 3.0 database.

### 2.8. Quantitative Verification of Differentially Expressed Genes

To further verify the differentially expressed genes of *B. pervariabilis* × *D. grandis* and *A. phaeospermum* obtained by dual-seq, 13 *B. pervariabilis* × *D. grandis* and 10 *A. phaeospermum* differentially expressed gene primers at four different infection stages were detected by qPCR (Appendix A). The nicotinamide adenine dinucleotide phosphate (NADPH) gene of *B. pervariabilis* × *D. grandis* and the NADPH gene of *A. phaeospermum* were selected as internal reference genes. Data were analyzed by the 2^−ΔΔCt^ method [45].

### 2.9. Screening and Validation of Effectors in A. Phaeospermum

Screening and validation of differentially expressed effectors at different stages of infection in *A. phaeospermum* were performed based on dual-seq sequencing, and the effectors of *A. phaeospermum* were predicted by using SignalP4.0, TargetP1.1, and TMHMM 2.0 software [46,47]. The enzyme digestion sites of four candidate effectors were analyzed by NEBcutter (http://tools.neb.com/NEBcutter2/index.php. Accessed on 1 September 2022). Primers containing the complementary sequence of the PGR-107 vector ClaI enzyme digestion site were designed using Primer 5.0 software (Appendix A), and the transient expression vector PGR-107 was linearized by ClaI enzyme (NEB) digestion. Reaction system: PGR-107 plasmid DNA 10 μL, 10× FlyCut Buffer 5 μL, ClaI 1 μL, and dd H_2_O 34 μL. Fifteen min at 37 °C and 20 min at 65 °C after mixing, agarose gel electrophoresis after gel recovery. The amplification effector was amplified from fungal cDNA with primers containing complementary sequences. Reaction system: DNA 1 µL, F/R 1/1 µL, and 2× TransTaq HiFi PCR SuperMix 25 µL nuclease-free water 22 µL. The reaction procedure was as follows: 94 °C for 5 min (94 °C for 30 s, 55 °C for 30 s, and 72 °C for 2 min) and 72 °C for 10 min. PCR products were detected by 1% agarose gel electrophoresis and recovered by gel cutting. The linearization vector PGR-107 was connected to the effector by a ClonExpress one-step cloning kit (Vazyme Biotech Co., Ltd. Nanjing, China). Reaction system: Linearized vector PGR-107 5 μL, effector DNA 3 μL, 5× CE Buffer 4 μL, Exnase 2 μL, and ddH_2_O 6 μL. After reaction at 37 °C for 30 min, the recombination reaction was completed by cooling at 4 °C for 5 min. The recombinant product was transformed by Stabl3 *E. coli*. The system: Connection product in ice bath for 25 min, 42 °C for 45 s, 2 min in ice bath, and add 500 μL LB, 200 rpm, 37 °C for 1 h. The coated plate was cultured upside down overnight. A single positive clone was picked, and 1–5^TM^ 2× High-Fidelity Master Mix was used for amplification by PCR and detection by 1% agarose gel electrophoresis. The plasmid DNA was sequenced. One microgram of recombinant plasmid was added to 100 μL of Agrobacterium competent GV3101 cells after mixing in an ice bath for 5 min, liquid nitrogen for 5 min, a 37 °C water bath for 5 min, and then an ice bath for 5 min. LB medium (900 μL) was added, and the cells were cultured at 28 °C and 200 rpm for 1 h. The collected bacteria were coated on 50 μg/mL kana LB plates and inverted for culture at 28 °C for 2–3 days. The positive cloned plasmid DNA was amplified by PCR using EmeraldAmp MAX PCR Master Mix (Takara, Dalian, China). Reaction system: Plasmid DNA 1 μL, EmeraldAmp MAX PCR Master Mix 25 μL, pGR-107F/R 1/1 μL, and ddH_2_O 22 μL. Procedure: 98 °C for 10 s, 60 °C for 30 s, 72 °C for 1 min, and 25 cycles in total. The PCR products were sequenced. The correctly sequenced Agrobacterium was selected and cultured in LB medium with 100 μg/mL Rif and 100 μg/mL kana, and the bacteria were collected after shaking culture at 28 °C and 220 rpm. The bacteria were suspended in MES buffer to make Agrobacterium with an OD600 = 0.5. Using a needle-free syringe, Agrobacterium containing the pGR-107 effector recombinant vector was injected into tobacco to avoid large leaf veins. Bax Agrobacterium and pGR-107 empty vector *Agrobacterium* were used as the positive control and negative control, respectively. The plates were incubated at 25 °C for 5 days under light for 16 h and dark for 8 h. The hypersensitivity reaction of tobacco was observed. Meanwhile, trypan blue staining [48] was used to identify cell death in tobacco.

### 2.10. Expression of Effectors in Tobacco Detected by RT–PCR

Healthy tobacco leaves and tobacco samples treated with the pGR-107 effector recombinant vector, Bax, and pGR-107 empty vector for 5 days were collected and ground after flash freezing in liquid nitrogen. Total RNA of tobacco was extracted with TransZol Plant (TransGen, Beijing, China), and cDNA was obtained by reverse transcription with All-in-One First-Strand cDNA Synthesis SuperMix for PCR (TransGen Biotech, Beijing, China). The tobacco Actin gene was used as an internal reference gene, and 2×TransTaq HiFi PCR SuperMix (TransGen Biotech, Beijing, China) was used for PCR amplification to detect the expression of effectors in tobacco.

### 2.11. Gene Knockout and Genetic Transformation

Previous studies have shown that *A. phaeopermum* cannot grow in 350 µg/mL hygromycin and 100 µg/mL geneticin, so it is used as a resistance marker gene for gene knockout and knockout complement. An improved split-marker method was used to identify gene function [49]. Taking the CDS region of *ApCE22* and *ApCE12* as the center, 1100 bp upstream and 1100 bp downstream were selected as homologous arms to design primers to amplify homologous arms of *ApCE22* and *ApCE12* from *A. phaeospermum* DNA. The sequence complementary to the upstream primer of the *hph* gene was added at 5′ of upstream primers, and the sequence complementary to the downstream primer of the *hhp* gene was added at 5′ of downstream primers. When the gene knockout complementation test was carried out, the sequences complementary to the upstream primers of the *kanMx* gene were added to the downstream primers of the *ApCE22* gene and *ApCE12* gene, and then the *ApCE22* and *kanMx* and *ApCE12* and *kanMx* genes were connected by fusion PCR. Then, the sequence complementary to the upstream primer of the *ApCE22* or *ApCE12* gene was added at 5′ of the ApCE22/ApCE12-5 downstream primers, and the sequence complementary to the downstream primer of *kanMx* was added at 5′ of the ApCE22/ApCE12-3 upstream primers. The primer sequences are shown in Appendix A. 2×TransTaq HiFi PCR SuperMix (TransGen Biotech, Beijing, China) was used to amplify the hygromycin phosphotransferase gene (*hph*) from the pSilent-1 vector and the *KanMx* gene from PUG6 plasmid DNA, ApCE22-5, ApCE22-3, ApCE12-5, and ApCE12-3 *hph*, and genetic genes, and *ApCE22* and *ApCE12* gene fragments were obtained. The fusion fragments of ApCE22-5+hph, hph+ApCE22-3, ApCE12-5+hph, hph+ApCE12-3, ApCE22+KanMx, ApCE12+KanMx, ApCE22-5+ApCE22+KanMx, ApCE22+KanMx+ApCE22-3, ApCE12-5+ApCE12+KanMx, and ApCE12+KanMx+ApCE12-3 were obtained by fusion PCR using LA Taq (Takara). All the fragments were subjected to agarose gel electrophoresis, and then the gel was recycled. *A. phaeopermum* protoplasts were prepared by lysine and Driselase (SIGMA) according to a previously described method [49]. Then, 30 μL of the fusion fragment ApCE22-*5*-hph/hph-ApCE22-*3* and ApCE12-*5*-hph/hph-ApCE12-*3* was added to the 5 × 10^6^ protoplast and incubated at room temperature for 20 min. Similarly, PEG-mediated protocol transformation was performed according to the same method as above. In the transformation detection of knockout strains, lysis buffer for micro-organization to direct PCR (TaKaRa, Dalian, China) was used to lyse the mycelium, and Ex Taq was used for detection. hph-F/R, (ApCE22-3+hph)-F/R, (ApCE12-5+hph)-F/R, ApCE22-F/R, and ApCE12-F/R were used for detection. The mutant strains were knocked out and supplemented, and 30 μL of the fragments ApCE22+KanMx/ApCE12+KanMx-5, ApCE22/ApCE12+KanMx, and ApCE22+KanMx/ApCE12+KanMx-3 was added to 5 × 10^6^ protoplasts and incubated at room temperature for 20 min. Similarly, PEG-mediated protocol transformation was performed using the above method [50]. In the transformation detection of knockout complementary strains, lysis buffer for micro-organization to direct PCR (TaKaRa, Dalian, China) was used to lyse the mycelium, and Ex Taq was used for detection. ApCE22+KanMx-F/ApCE22+KanMx-R/ApCE12+KanMx-F/ApCE12+KanMx-R were used for detection. The primers are shown in Appendix A.

### 2.12. Phenotypic Analysis and Pathogenicity Test of Mutant and Complementary Strains

To analyze the differences in vegetative growth among the strains, including gene knockout strains and gene knockout complementary strains, stress sensitivity assays were conducted on PDA plates supplemented with different agents: 2 mg/mL Congo red (CR), 2 mol/L NaCl, and 40 mmol/L H_2_O_2_ at 25 °C for 5 days. All assays were repeated three times, and all data were analyzed by one-way ANOVA and Duncan’s range test in SPSS 16.0 to measure specific differences between pairs of means. A *p* value of <0.05 was considered statistically significant.

Thirty plants of two-year-old *B. pervariabilis* × *D. grandis* with uniform growth were selected. Five plants were randomly selected and sprayed with a mycelial suspension of the wild-type strain on the upper eight shoots of each plant. The remaining 25 plants were treated with the mycelial suspension of mutant Δ*ApCE12*, the mycelial suspension of mutant Δ*ApCE22*, the mycelial suspension of the *ApCE12* complementary strain, the mycelial suspension of the *ApCE22* complementary strain, and sterile water, with five plants per treatment. The samples were subjected to bagging moisturizing and sprayed once every 12 h, 3 times in total, using 3 independent replicates. The incidence was investigated 20 days after inoculation. The disease index was calculated as follows [51]. Disease grading standard: Grade 0: no wilt; Grade 1: less than 25% branches withered; Grade 2: 25–50% (including 25% and 50%) branches withered; Grade 3: 50–75% of the branches dead (including 75%); Grade 4: more than 75% branches withered. Disease index = [Σ (disease grade × number of diseased branches)/(total branches) × the most serious disease grade] × 100.

## 3. Results

### 3.1. Sequencing Data Quality Detection and Alignment to the Reference Genome

A total of 189.22 G of raw data were obtained by dual-seq sequencing. The sequencing results were filtered to remove low-quality reads, and 181.18 G of clean data were obtained. The quality test results are shown in Table 1. The filtered clean reads were compared to the reference genomes of *A. phaeospermum* and *B. pervariabilis* × *D. grandis*. *A. phaeospermum* was compared to the whole genome (Table 2), and hybrid bamboo was compared to the assembled reference genome (Table 3). In S1, before *A. phaeospermum* infection of *B. pervariabilis* × *D. grandis*, almost no sequences were aligned to the *A. phaeospermum* reference genome. With the increase in the *A. phaeospermum* infection time of *B. pervariabilis* × *D. grandis*, the number of sequences aligned to the *A. phaeospermum* reference genome in the dual-seq sequencing results increased. At the later stage of S4 infection, 50.2% of the sequences could be compared to the *A. phaeospermum* reference genome. In S1 and S2 before *A. phaeospermum* infection of *B. pervariabilis* × *D. grandis*, there was no significant difference in the number of reads compared to the reference genome of *B. pervariabilis* × *D. grandis*, which was higher than 70%. With the increase in infection time, the number of reads compared to the reference genome of *B. pervariabilis* × *D. grandis* gradually decreased, and in S4, the number of reads compared to the reference genome of *B. pervariabilis* × *D. grandis* was as low as 36.16%. All interaction transcriptome data of 12 samples of *B. pervariabilis* × *B. grandis* were deposited in the national center for biotechnology information (NCBI) Sequence Reads Archive (SRA) under the accession numbers SRR14685222, SRR14685221, SRR14685220, SRR14685219, SRR14685218, SRR14685217, SRR14685216, SRR14685215, SRR14685214, SRR14685213, SRR14685212, and SRR14685211. The raw data were published in the NCBI Sequence Read Archive (SRA) under the Bio Project accession number SAMN19312317 (https://www.ncbi.nlm.nih.gov/biosample/?term=SAMN19312317, accessed on 23 May 2021).

### 3.2. Dual-seq Differentially Expressed Gene Analysis

The differentially expressed genes of *A. phaeospermum* and hybrid bamboo in the interaction samples of S1, S2, S3, and S4 were analyzed. In the analysis of differentially expressed genes of hybrid bamboo in four periods, it was found that a total of 40 differentially expressed genes were obtained in six comparison groups (Figure 1). The results of the cluster analysis show that the expression levels of these differentially expressed genes increased with the increase in the infection process. At the S4 stage, these differentially expressed genes were all at a high level of expression (Figure 2). In addition, there were 1891, 1172, and 29,098 differentially expressed genes in the S2 vs. S1, S3 vs. S1, and S4 vs. S1 comparison groups, respectively, while there were 1629, 31,900, and 22,506 differentially expressed genes in the S3 vs. S2, S4 vs. S2, and S3 vs. S4 comparison groups, respectively (Figure 3). In addition, the gene expression of pathogenic bacteria in the three infection stages of S2, S3, and S4 was also different. There were 3098, 2868, and 7296 differential genes in S3 vs. S2, S4 vs. S2, and S3 vs. S4, respectively. The volcanic map of differential genes in these three comparison groups is shown in Figure 4.

### 3.3. GO and KEGG Enrichment Analyses of Differentially Expressed Genes

Compared with S1, the differentially expressed genes of *B. pervariabilis* × *D. grandis* in S2, S3, and S4 in response to *A. phaeospermum* infection were mainly enriched in biological processes, such as response to stress, DNA metallic processes, aminoglycan catabolic processes, regulation of innate immune responses, carbohydrate metallic processes, oxidation reduction processes, single organization metallic and other biological processes, as well as catalytic activity, oxidoreductase activity, intracellular, nonmembrane-bound organelles, cytoskeletal parts, other molecular functions, carbohydrate derivative binding, adenyl nucleoside binding, nucleoside binding, pyrophagase activity and pure ribonucleoside binding, apoplasts, adenosine diphosphate (ADP) binding, and other cellular components. In addition, the differentially expressed genes in the S3 group vs. S2 group are mainly enriched in biological processes, such as response to oxidative stress and response to peroxide, molecular functions, such as cytoskeleton construction, and molecular functions, such as catalytic activity, oxidoreductase activity, and hydrolase activity. The S4 group vs. S2 group is mainly enriched in biological processes, such as carbohydrate metabolism and redox, cell wall, and extracellular synthesis, as well as molecular functions, such as catalytic activity, oxidoreductase activity and hydrolase activity. The S4 group vs. S3 group is mainly enriched in biological processes, such as metabolism, redox and carbohydrate metabolism of a single organism, molecular components, such as transcription factor complexes and cell wall composition, as well as molecular functions, such as catalytic activity, oxidoreductase activity, and hydrolase activity. In the process of interaction between *A. phaeospermum* and hybrid bamboo, the differentially expressed genes in the process of *A. phaeospermum* infecting *B. pervariabilis* × *D. grandis* in S3 and S4 were mainly enriched in biological processes, gene expression, multicellular organic processes, organic substance metallic processes, cellular metallic processes, macromolecule metabolic process, and other biological processes, as well as intracellular, cell macromolecular complex, and other cellular components, in addition to binding, oxidoreductase activity, substrate specific transporter activity, and other molecular functions. The GO enrichment pathways of the *B. pervariabilis* × *D. grandis* differentially expressed genes in six comparison groups (S4 vs. S2, S4 vs. S1, S4 vs. S3, S3 vs. S2, S3 vs. S1, and S2 vs. S1) are shown in Figure 5. In addition, the GO enrichment pathways of the *A. phaeospermum* differentially expressed genes in three comparison groups (S4 vs. S2, S4 vs. S3, and S3 vs. S2) are shown in Figure 6.

The results of the KEGG enrichment analysis show that the differentially expressed genes in *B. pervariabilis* × *D. grandis* were significantly enriched in sucrose metabolism, phenoid biosynthesis, cutin, suberine, wax biosynthesis, plant hormone signal transduction, and plant–pathogen interaction. Compared with S1, 16, 18, and 210 differentially expressed genes of *B. pervariabilis* × *D. grandis* in the S2, S3, and S4 stages were enriched in the plant hormone signal transduction pathway, respectively. In addition, 20, 10, and 233 differentially expressed genes were enriched in the metabolic pathway of phenylpropanoid biosynthesis. In addition, 10, 7, and 136 differentially expressed genes were enriched in plant–pathogen interactions. The differential gene KEGG enrichment pathways of six comparison groups (S4 vs. S2, S4 vs. S1, S4 vs. S3, S3 vs. S2, S3 vs. S1, and S2 vs. S1) are shown in Figure 7. The KEGG enrichment pathways of the *A. phaeospermum* differentially expressed genes in three comparison groups (S4 vs. S2, S4 vs. S3, and S3 vs. S2) are shown in Figure 8. The differentially expressed genes in the S4 group vs. S2 group are mainly enriched in KEGG enrichment pathways, such as amino sugar and nucleotide sugar metabolism, starch and sucrose metabolism, ribosomes, phenylpropionic acid biosynthesis, keratin, cork alkali and wax biosynthesis, citric acid cycle, and plant–pathogen interaction. The differentially expressed genes in the S4 group vs. S3 group are mainly enriched in KEGG enrichment pathways, such as starch and sucrose metabolism, phenylalanine biosynthesis, cysteine and methionine metabolism, amino sugar and nucleotide sugar metabolism, plant–pathogen interaction, citric acid cycle, horniness, cork alkaloid, and wax biosynthesis. The differentially expressed genes in the S3 group vs. S2 group are mainly enriched in the KEGG enrichment pathways, such as endoplasmic reticulum protein processing, starch and sucrose metabolism, plant hormone signal transduction, phenylpropionic acid biosynthesis, amino sugar and nucleotide sugar metabolism, keratin, cork alkali and wax biosynthesis, and plant circadian rhythm.

In the differential analysis of the pathogen, it was found that the gene cutinase transcription factor closely related to the breakthrough of the host bamboo epidermis was heavily expressed in the S2 period. With the increase or decrease in the infection time, the expression of this gene was significantly reduced in the S3 infection period compared with the S2 period, while the S4 infection period was significantly reduced compared with the S3 period. It was analyzed that the pathogen was highly expressed in the S2 period just contacting the host hybrid bamboo epidermis to help the pathogen break through the epidermal invasion, The downregulation in S3 and S4 may be since other pathogenic factors, such as effectors, are needed to help the pathogen colonize and multiply to further infect the host after the pathogen invades the hybrid bamboo, rather than the function of related genes, such as cutinase, that help break through the epidermal barrier, so the expression decreases. However, in the analysis of differentially expressed genes of S2 vs. S1, it was found that these differentially expressed genes were mainly enriched in the PAMP-triggered immunity process caused by the pathogenic fungus race-specific-elicitor A9 (Avr9). When the race-specific-elicitor A9 (Avr9) of *A. phaeospermum* contacts the PRR disease resistance protein Cf9 on the plant surface, it causes the response of calcium-dependent protein kinase in the cytoplasm, causing the response of respiratory burst oxidation and producing a large number of ROS through the action of NADPH oxidation, resulting in the PAMP-triggered immunity of a hypersensitive response. In the differential gene analysis of S2 vs. S1, no immune response caused by fungal effectors was found. By analyzing the differential genes of S3 vs. S1 in this interaction pathway, it was found that the resulting immune response was consistent with that of S2 vs. S1. When the differentially expressed genes in S4 vs. S1 were enriched in the plant–pathogen interaction pathway (Appendix A), it was found that the fungal effector AVRA10 protein avoided the recognition of PRRS on the plant surface and entered the plant cytoplasm, causing the response of the R protein MAL10, resulting in the differential expression of the downstream gene WRKY transcription factor. Finally, it made the host plant have defense-related gene induction and programmed cell death. These differentially expressed genes respond to fungal effector-triggered immunity. In addition, it was also found that the differential expression of cyclic nuclear-gated channel genes on the plant surface led to the simultaneous response of calmodulin, calcium-dependent protein kinase, and respiratory burst oxidase to invasion after external calcium ions entered the cytoplasm. Through the action of nitric oxide synthase and NADPH oxidase, a large number of ROS and NO were produced, which reinforced the plant cell wall and stomatal closure and produced a hypersensitive response.

### 3.4. Dual RNA-seq Interaction Gene Annotation

Comparing the gene annotation results of *A. phaeospermum* in the correlation analysis with the STRING database, 2987 pairs of protein interactions were obtained. A total of 3642 and 2656 genes were annotated in the HPIDB 3.0 database and VFDB database, respectively, including 114 virulence factors. A total of 441 genes were annotated in the HPIDB 3.0 database, and 53 pairs of genes interacted between *A. phaeospermum* and *B. pervariabilis* × *D. grandis*, 8 pairs of which were direct interactions (Appendix A). Among the 5 pairs of interaction relationships, 3 genes annotated in the PHI database (AP-Z1314754, AP-Z1317911, and AP-Z1319361) were involved. In addition, there were 4 genes (AP-Z1319361, AP-Z1314754, AP-Z1310389, and AP-Z1317955) annotated in the VFDB database, and these genes are considered to be pathogenic genes, virulence genes, and effector genes of pathogens. However, their direct interaction genes in bamboo were mainly annotated with the thioredoxin M type, heat shock protein family, ubiquitin receptor, xylanase inhibitor, and apoptosis-related genes in the Arabidopsis database. Most of these genes are involved in the immune response of plants to external stress. Therefore, the genes interacting with these virulence factors and effectors in *B. pervariabilis* × *D. grandis* are considered to be immune response genes and resistance genes in plants in response to biological stress.

### 3.5. Prediction of Effectors of A. Phaeospermum

To predict the effector protein, SignalP4.1 was used to predict the signal peptide; the predicted signal peptide protein was selected; TargetP1.1 was used to predict the subcellular localization; the protein localized to S was selected, and TMHMM2.0 was used to predict the transmembrane domain of the protein. Finally, 256 candidate effectors of *A. phaeospermum* were predicted, including five new genes. By analyzing the differentially expressed genes of S3 vs. S2, S4 vs. S2, and S4 vs. S3, it was found that there were 33, 33, and 89 differentially expressed candidate effectors, respectively. Compared with S3 vs. S2, there were 59 unique differentially expressed candidate effectors in the two comparison groups of S4 vs. S2 and S4 vs. S3. Coincidentally, the differentially expressed gene analysis of hybrid bamboo in response to *A. phaeospermum* infection in the four S1–S4 stages shows that in S3 and S2, *A. phaeospermum* only triggered the immune response caused by PAMP in *B. pervariabilis* × *D. grandis*. The immune response triggered by *A. phaeospermum* candidate effectors in *B. pervariabilis* × *D. grandis* was significantly stimulated in the S4 period. This result shows that 59 unique differentially expressed candidate effectors in S4 may play a key role in stimulating the ETI of *B. pervariabilis* × *D. grandis*. Fifty-three differentially expressed candidate effectors were BLAST-compared and annotated in the nonredundant protein sequence database (Nr), SwissProt, GO, and KEGG databases. Three candidate effectors, *ApCE12*, *ApCE22*, and *ApCE28*, with unknown gene functions were obtained, and the other proteins with specific functions were significantly enriched in the biosynthesis of secondary metabolites, inositol phosphate metabolism, microbial metabolism in diverse environments, and genetic information processing.

### 3.6. qRT–PCR Verification of Differentially Expressed Genes in Dual-seq Sequencing

To verify the sequencing results of dual-seq sequencing, RT-qPCR was used to detect the expression differences of 13 shared differentially expressed genes of *B. pervariabilis* × *D. grandis* in six comparison groups (S3 vs. S1, S2 vs. S1, S4 vs. S1, S3 vs. S2, S4 vs. S2, and S4 vs. S3) and 10 shared differentially expressed genes in three comparison groups (S3 vs. S2, S4 vs. S2, and S4 vs. S3). The NADPH of *B. pervariabilis* × *D. grandis* and the NADPH of *A. phaeospermum* were selected as internal reference genes. The qPCR results are shown in Figure 9. The fluorescence quantitative verification results of these differential genes are almost consistent with the transcriptional sequencing results, which proves that the dual-seq sequencing results are reliable.

### 3.7. Tobacco Transient Expression Identification Effector

Using the pGR107 vector as the backbone vector, the tobacco transient expression vector pGR107-ApCEs was constructed for three candidate effectors (*ApCE12*, *ApCE22*, and *ApCE28*) that were not functionally annotated and could trigger the hybrid bamboo ETI response. Sequencing using the primers of the pGR107 vector was correct. The correctly sequenced recombinant plasmid DNA was transferred into Agrobacterium, cultured to an OD600 of 0.8, and then inoculated into tobacco leaves by osmosis for 5 days. The tobacco symptoms are shown in Figure 10. *ApCE*s could be detected in the injected area in *N. benthamiana*. The transient expression results show that *ApCE22* effectors caused obvious programmed cell death in tobacco leaves, while *ApCE12* caused slighter cell death in tobacco leaves. In addition, trypan blue staining showed that tobacco cells inoculated with Bax protein at effectors ApCE12 and ApCE22 turned blue, while cells inoculated with sterile water did not turn blue. It indicates that effectors *ApCE12* and *ApCE22* cause programmed cell death in tobacco. It is speculated that they may be a protein toxin or an effector that can be recognized by tobacco.

### 3.8. Results of Gene Knockout and Knockout Complementation

#### 3.8.1. Amplification of the Fusion Fragment and Genetic Transformation of Mutant and Complementary Strains

Gene fragments in the construction of the knockout vector and knockout complement vector were successfully constructed by an improved split-marker method. The fusion fragments of ApCE12-5+hph, hph+ApCE12-3, ApCE22-5+hph, and hph+ApCE22-3 obtained by two rounds of fusion PCR and the knockout complementation fusion fragments ApCE22-5+ApCE22+KanMx and ApCE12-5+ApCE12+KanMx are shown in Appendix A. The mycelium of wild-type A. phaeospermum and mutants Δ*ApCE12* and Δ*ApCE22* were hydrolyzed with 0.2 g lysine and 0.5 g Driselase, respectively. In addition, a large number of protoplasts were obtained, which were swollen and transparent at a dilution concentration of 10^7^/mL (Appendix A). Then, the fusion fragment was transferred into the protoplast and cultured in TB3 medium for 3 days. A single colony was transferred to a PDA plate with a hygromycin concentration of 350 µg/mL, and several transformants were obtained after 4 days of culture. Then, the “*hph*”, “*ApCE12*”, “*ApCE22*”, and *KanMx* fragments of the wild-type, *ApCE22*, *ApCE12* deletion mutant strain, *ApCE22* complementary strain, and *ApCE12* complementary strain were detected by PCR, as shown in Appendix A. The sequencing results are consistent with the previous results.

#### 3.8.2. Phenotypic Analysis of Transformants

The phenotypes of the mutants were significantly different from those of the wild-type and complementary strains (Figure 11). Based on the percent growth inhibition of the strains relative to unstressed controls, the relative growth inhibition of Δ*ApCE12* and Δ*ApCE22* colonies was 84.5% and 76.5% in the experiment adding 40 mmol/L H_2_O_2_, respectively. Both mutants were significantly more sensitive to the oxidative stress of H_2_O_2_ than the wild-type strains (52.6%) compared to the estimates from the wild-type strain. Moreover, two mutants exhibited significantly decreased tolerance to NaCl. Δ*ApCE12,* and Δ*ApCE22* drastically decreased the colony growth areas by 70% and 56.4% at a concentration of 2 mg/mL NaCl, respectively. However, the stress of 2 mol/L Congo red caused no significant differences in colony growth in either of the disruption mutants. There was no significant difference in phenotype between the wild-type and complementary strains (Figure 12).

#### 3.8.3. Pathogenicity Test of Transformants

To determine whether Δ*ApCE12* and Δ*ApCE22* are involved in pathogenicity, we performed a pathogenicity test on twigs and leaves by inoculating mycelial plugs of the wild type, Δ*ApCE12* deletion mutant, and Δ*ApCE22* deletion mutant. Mild symptoms were found on the Δ*ApCE12* and Δ*ApCE22* mutant-infected twigs after culture in a greenhouse, while obvious symptoms on twigs and leaves of *B. pervariabilis* × *D. grandis* were found in the wild-type strains, *ApCE12* complementary strain Δ*ApCE12*+, and *ApCE22* complementary strain Δ*ApCE22*+ (Figure 13). The disease index of *B. pervaria-bilis* × *D. grandis* inoculated with sterile water did not change with increasing time and was 0. The disease index of *B. pervariabilis* × *D. grandis* inoculated with the wild type, mutant Δ*ApCE12*, mutant Δ*ApCE22*, Δ*ApCE12*+, and Δ*ApCE22*+ increased significantly with increasing inoculation time; after 25 days of inoculation, the disease index reached 80%, 52.5%, 47.5%, 77.5%, and 75%, respectively. At the same time, we found that the disease index of *B. pervariabilis* × *D. grandis* inoculated with the wild type, Δ*ApCE12*+, and Δ*ApCE22*+ was significantly higher than that inoculated with the mutant Δ*ApCE12* strain and mutant Δ*ApCE22* strain at the same time point. However, there was no significant difference in the disease index between the mutants Δ*ApCE12* and Δ*ApCE22*. (Figure 14).

## 4. Discussion

The results of the interaction analysis between hybrid bamboo and *A. phaeospermum* show that compared with uninfected hybrid bamboo, the differentially expressed genes of hybrid bamboo in early stage S2, midinfection stage S3, and late infection stage S4 were significantly enriched in plant–pathogen interaction, plant hormone signal transduction, glutathione metabolism, and phenylpropanoid biosynthesis. These results suggest that these metabolic pathways in hybrid bamboo may play an important role in its immune response. In addition to hybrid bamboo, these metabolic pathways have been proven to be related to disease resistance in a variety of plants. Glutathione metabolism and plant hormone signal transduction in *Gossypium barbadense* respond positively after being infected by *Fusarium oxysporum* to resist the invasion of pathogens [52]. The differentially expressed genes of Arabidopsis infected by *Verticillium dahliae* are mainly enriched in metabolic pathways, such as plant–pathogen interaction and plant hormone signal transduction [53]. After the wheat was infected by *Fusarium pseudograminearum*, phenylpropane biosynthesis, glutathione metabolism, and plant hormone signal transduction occurred in defense [54].

In the immune response of hybrid bamboo to *Arthrodia obscura*, the differentially expressed genes in the plant–pathogen interaction pathway in the early S2 stage and middle S3 stage of infection were more abundant than those in hybrid bamboo before infection, mainly calcium-dependent protein kinases (CDPKs), calmodulin (CALM), respiratory burst oxide (RbOH), heat shock protein 90 kDa beta (HSP90), pathway-related gene transcriptional activator (PtI6) and pathway-related protein 1 (PR1). CDPKs and CALM are important calcium sensor proteins. When attacked by pathogens, Ca^2+^ signal transduction occurs in plants and activates the plant’s immune response [55]. For example, when mildews infect barley, calmodulin binds to MLO-resistant proteins to regulate the defense response of barley to molds [56]. RbOH protein can produce ROS after plants recognize the pathogens and resist the invasion of pathogens by activating hormones, such as salicylic acid (SA), ethylene, and jasmonic acid [57]. In addition, ROS play an important role in the resistance of potato to *Phytophthora infestans* [58], tomato to *Xanthomonas campestris* [59], and *Arabidopsis thaliana* to *Pseudomonas* [60]. HSP90 is an important molecular chaperone in plants that can mediate the immune response of cassava bacterial bright and increase its disease resistance [61]. Previous studies have shown that the expression of the tomato transcription factors PTI4, PTI5, and PTI6 is enhanced in response to *Pseudomonas syringae* pv infection, and they can also mediate the expression of salicylic acid, ethylene, and jasmonic acid-regulated pathogenesis-related protein 1, thereby increasing the disease resistance of host plants [62,63]. These differentially expressed genes are mainly involved in plant signal transduction and defense-related gene induction after pathogen invasion and participate in plant PTI triggered by pathogen PAMP.

However, in the comparison of the later S4 stage of infection compared with the previous three stages (S1, S2, and S3), the differentially expressed genes enriched in the plant–pathogen interaction pathway include cyclic nucleus gated channel (CNGC), mi-togen-activated protein kinase kinase kinase kinase 1 (MEKK1), mitogen-activated pro-tein kinase kinase kinase 4/5 (MKK4/5), mitogen-activated protein kinase (MAPK), WRKY transcription Factor 1/25/29 (WRKY1/25/19), resistance protein RP2 (RP2), and the suppressor of G2 allele of SKP1 (SGT1). CNGC is a cation channel for plants to transmit calcium ions. The CNGC gene family in tomato is closely related to the immune response triggered by its own PAMP. In tomato CNGC gene-silencing plants, the immune response triggered by PAMP is significantly weakened, indicating that CNGC plays an important role in plant disease resistance [64,65]. MAPK cascades are a part of PTI signal transduction in plants. Among the differentially expressed genes in S4 com-pared with the previous three periods, there is an important MAPK cascade (MEKK1, MKK4, and MKK5), which can be activated by self-defense-related genes and pathogens. This MAPK cascade has been proven to play an important role in Arabidopsis innate immunity, in which MKK4 and MKK5 are closely related to apoptosis [66,67]. Therefore, we speculate that *A. phaeospermum* can activate the MAPK cascade reaction and trigger PTI in hybrid bamboo during S4 infection. WRKY transcription factors are involved in defending against the attack of plant pathogens. As a transcription factor that can activate ETI, it plays a key role in plant resistance to biological stresses, such as fungi. In addition, WRKY transcription Factor 1/25/29 is a unique differentially expressed upregu-lated gene in hybrid bamboo in the S4 period. Previous studies have found that WRKY can improve the resistance of Arabidopsis thaliana to *Botrytis cinerea* and *Alternaria Brassica* and the resistance of *Brassica rapa* ssp. *Pekinensis* to *Peronospora parasitica* [68,69]. In addition, the WRKY25 transcription factor in Arabidopsis has been proven to play an important role in defense against *P**. syringae* [70]; the WRKY1 transcription factor in grapes has been proven to play a role in response to pathogen infection, such as Pythium, downy mildews, and powdery mildews [71], and WRKY1 in wheat has been proven to play a role in response to powdery mildews [72]. The specific differential genes of hybrid bamboo in the S4 period also include resistance proteins, which can recognize the effectors triggering ETI in pathogens and increase plant disease resistance [73]. Therefore, in S2 and S3 after *A. phaeospermum* infection of hybrid bamboo, PTI of the host was caused, while ETI was activated in S4, indicating that *A. phaeospermum* infection of hybrid bamboo is a relatively slow process, which may be related to the high temperature during the infection period. The incidence rate of hybrid bamboo shoot blight is closely related to the ambient temperature; the lower the temperature is, the faster and more serious the disease.

Therefore, in S2 and S3 after *A. phaeospermum* infection of hybrid bamboo, only the PTI of the host hybrid bamboo occurs, while the ETI is activated in S4, indicating that the invasion of the fungus causing the hybrid bamboo to wilt is a relatively slow process. In addition, the results indicate that the fungal differentially expressed effectors unique to the S4 infection period may be key in triggering the ETI of hybrid bamboo, and the triggering of ETI is usually accompanied by HR cell death [74]. In this study, effectors that may cause tobacco hypersensitivity were identified by the tobacco transient expres-sion system. This finding indicates that the effectors *ApCE22* and *ApCE12* may be recog-nized by the tobacco immune system or could be virulence-related effectors. In addition, we obtained the mutants ΔApCE12 and ΔApCE22 of *A. phaeospermum* effectors *ApCE12* and *ApCE22* by gene knockout. In the determination of stress resistance of the mutants Δ*ApCE12* and Δ*ApCE22*, it was found that their sensitivity to hydrogen peroxide stress was significantly increased compared with the wild-type strain, indicating that the deletion of the effectors *ApCE12* and *ApCE22* may affect the metabolism of reactive oxygen species of *A. phaeospermum*. This result is consistent with the research result that the effector *Pep1* of *Ustilago maydis* suppresses plant immunity by disturbing the balance of active oxygen species of the host [75]. It indicates that when the wild-type *A. phaeospermum* invades the hybrid bamboo, the first immune defense of the host is rapidly activated, and reactive oxygen species are produced to resist the pathogen. The wild-type strain inhibits this immune process through the utilization and metabolism of reactive oxygen species. However, when the metabolism of reactive oxygen species of the mutant strains, no matter Δ*ApCE12* or Δ*ApCE22*, was reduced, the host hybrid bamboo could not inhibit its immune response due to the production of a large amount of reactive oxygen species. Therefore, the effectors ApCE12 and ApCE22 may play an important role in inhibiting plant PTI immunity, which is also the reason why the mutants Δ*ApCE12* and ΔApCE22 are less virulent than the wild type. We also found that there was no significant difference in the growth status among the wild type, ΔApCE12, and ΔApCE2 under Congo red stress, indicating that the effectors *ApCE12* and *ApCE22* did not affect the formation of the *A. phaeospermum* cell wall. *A. phaeospermum* is an advanced fungus with septal hyphae, and the main component of its cell wall is chitin, which contains only a small amount of cellulose [76]. It is well known that Congo red can combine with cellulose to form a red complex, which has a significant impact on the formation of chitin-based pathogen cell walls. In addition, the sensitivity of the mutants Δ*ApCE12* and Δ*ApCE22* to salt stress was significantly higher than that of the wild-type strain. This indicates that the effectors *ApCE12* and *ApCE22* may play an important role in the metabolic pathway of sodium chloride. Finally, compared with the wild-type, the disease index and symptoms of hybrid bamboo after being infected by mutants, no matter Δ*ApCE12* or Δ*ApCE12*, were significantly reduced. Therefore, *ApCE22* and *ApCE12* are virulence-related effectors that have a virulence function similar to that of the MC69 effector in *Magnaporthe oryzae*. They mainly destroy host cells in a necrotrophic infection mode, killing host tissue to support nutrition for self-growth [77,78]. Meanwhile, the degree of tobacco programmed death caused by *ApCE12* is significantly lower than that caused by *ApCE22*. It is speculated that ApCE12 can protect its pathogen from being recognized by the host plant to a certain extent, to help the pathogen penetrate the host plant and eventually lead to disease. However, the target genes of their interaction in the host hybrid bamboo plant are not clear. In the future, it is necessary to use yeast two-hybrid technology, pulldown hybridization, and two-way immunofluorescence technology to screen and verify the target genes of the hybrid bamboo effector *ApCE*s to comprehensively analyze the pathogenic mechanism of hybrid bamboo blight.

## 5. Conclusions

This research for the first time completes the sequencing analysis of differentially expressed genes in four periods (S1–S4) after *A. phaeospermum* infected the hybrid bamboo based on the dual RNA-seq. The expression of the effectors *ApCE12* and *ApCE22* were significantly upregulated in S4 period when the ETI reaction was triggered. It is speculated that they are related to the ETI reaction of hybrid bamboo. It was found that the effectors *ApCE12* and *ApCE22* were closely related to the virulence of *A. phaeosper**-mum*, but did not participate in the synthesis of the cell wall of *A. phaeospermum*. In addition, the effectors ApCE12 and ApCE22 were also found to play an important role in sodium chloride metabolism and could inhibit the immune response caused by the burst of reactive oxygen species in hybrid bamboo.

## Figures and Tables

**Figure 1 biomolecules-12-01264-f001:**
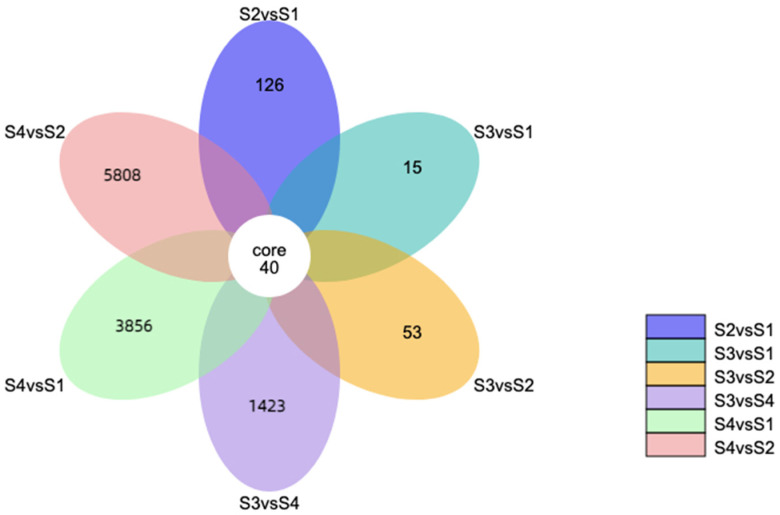
Venn diagram of 6 comparison groups in four periods S1–S4. Different colors represent different expressed genes in different comparison groups, and overlapping positions represent common different expressed genes in different comparison groups.

**Figure 2 biomolecules-12-01264-f002:**
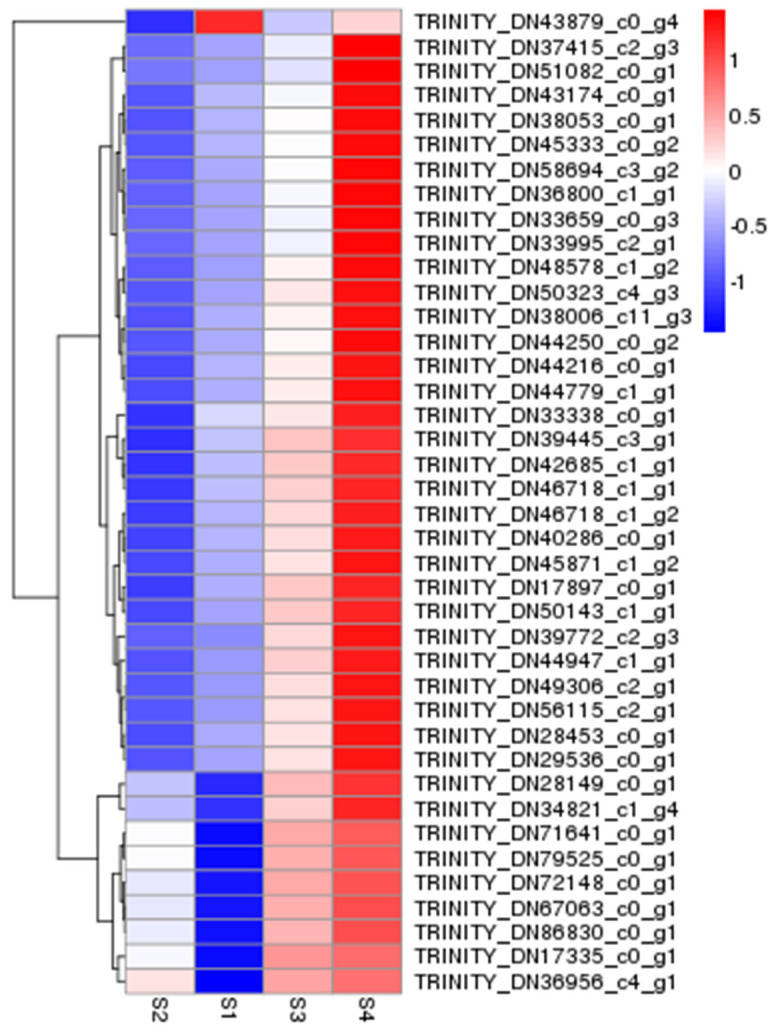
Clustering heatmap of 40 common differentially expressed genes in four periods S1–S4. S1–S4 represent different periods of *A. phaeospermum* infection *B. pervariabilis* × *D. grandis*. Blue represents low gene expression, and the darker the color, the lower the expression. Red represents high gene expression, and the darker the color, the higher the expression.

**Figure 3 biomolecules-12-01264-f003:**
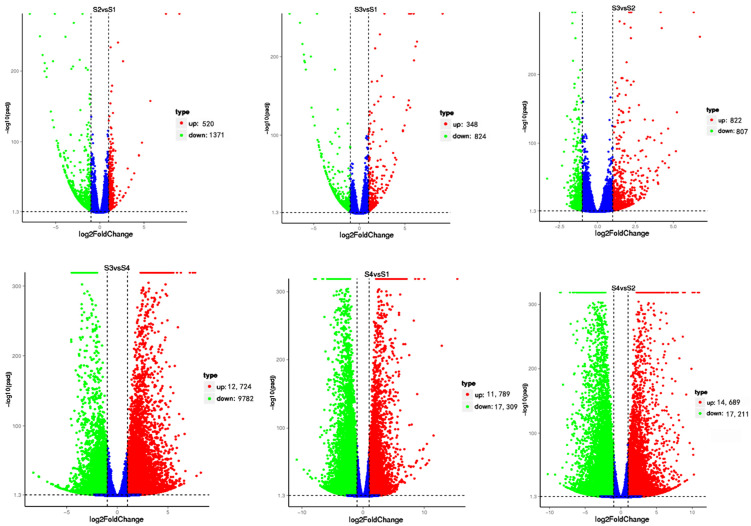
*B. pervariabilis* × *D. grandis* differentially expressed genes of 6 comparison groups in four periods S1–S4. Note: Red dots represent upregulated genes; green dots represent down regulated genes, and blue dots represent nondifferentially expressed genes.

**Figure 4 biomolecules-12-01264-f004:**
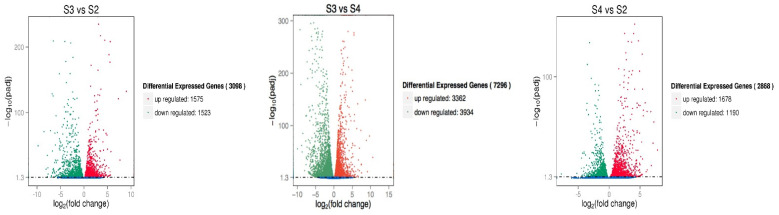
*A. phaeospermum* differentially expressed genes of 3 comparison groups in three periods S2–S4. Note: Red dots represent upregulated genes; green dots represent down regulated genes and blue dots represent nondifferentially expressed genes.

**Figure 5 biomolecules-12-01264-f005:**
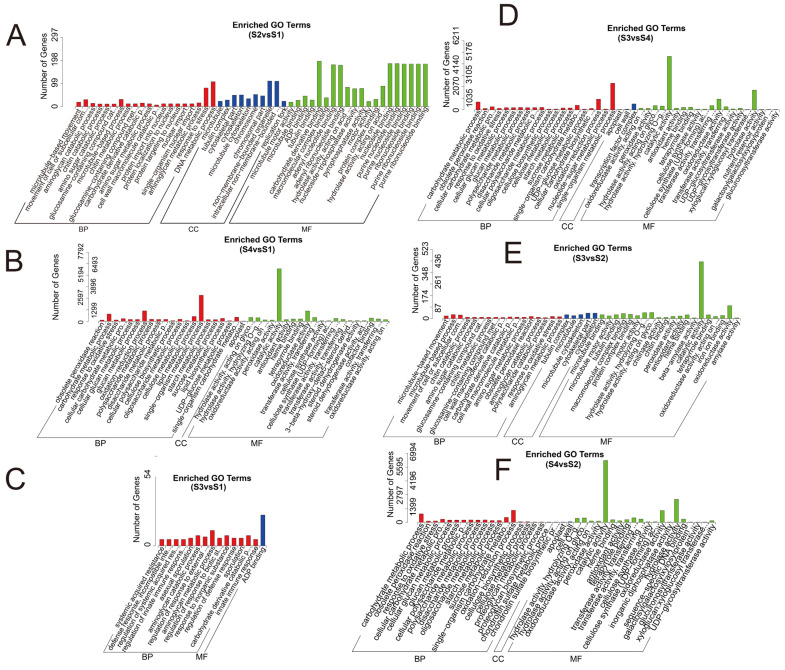
Go enrichment analysis of differentially expressed genes in *B. pervariabilis* × *D. grandis* in four periods S1–S4. Note: (**A**): S2 vs. S1; (**B**): S4 vs. S1; (**C**): S3 vs. S1; (**D**): S3 vs. S4; (**E**): S3 vs. S2; (**F**): S4 vs. S2. The red histogram represents biological processes; the blue histogram represents cellular components; the green histogram represents molecular functions.

**Figure 6 biomolecules-12-01264-f006:**
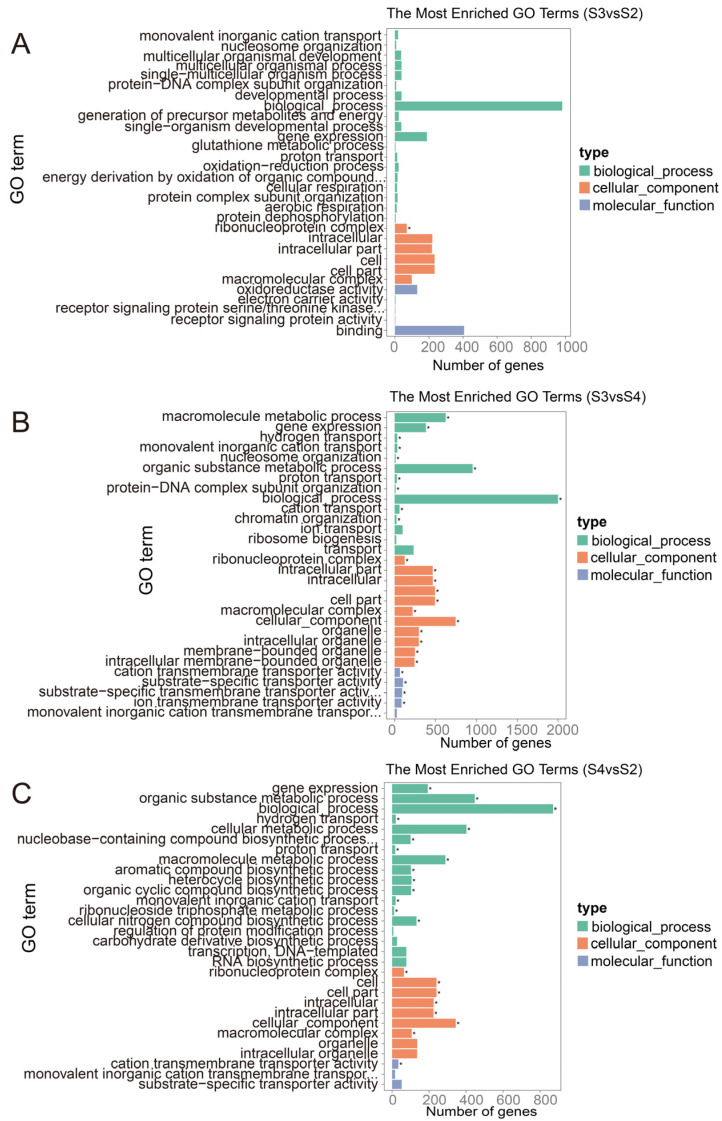
Go enrichment analysis of differentially expressed genes in *A. phaeospermum* in three periods S2–S4. (**A**): Go enrichment analysis of differentially expressed genes in *A. phaeospermum* of S3 vs. S2. (**B**): Go enrichment analysis of differentially expressed genes in *A. phaeospermum* of S3 vs. S4. (**C**): Go enrichment analysis of differentially expressed genes in *A. phaeospermum* of S4 vs. S2. Note: The green histogram represents biological processes; the orange histogram represents cellular components; the purple histogram represents molecular functions. "*" indicates enriched GOterm.

**Figure 7 biomolecules-12-01264-f007:**
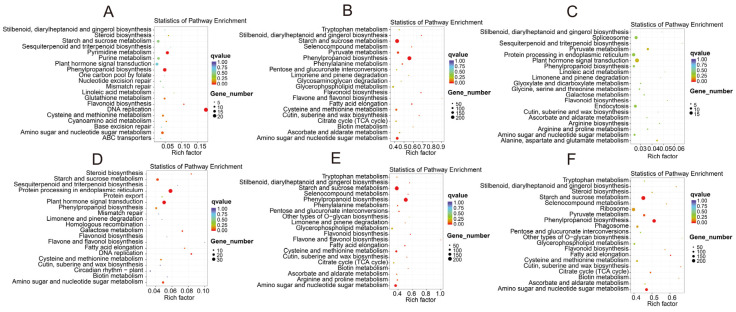
KEGG pathway analysis of differentially expressed genes in *B. pervariabilis* × *D. grandis* in four periods S1–S4. Note: (**A**): S2 vs. S1; (**B**): S4 vs. S1; (**C**): S3 vs. S1; (**D**): S3 vs. S2; (**E**): S3 vs. S4; (**F**): S4 vs. S2. Note: The color of the dot represents different qvalues, and the size of the dot represents the number of differentially expressed genes in this pathway.

**Figure 8 biomolecules-12-01264-f008:**
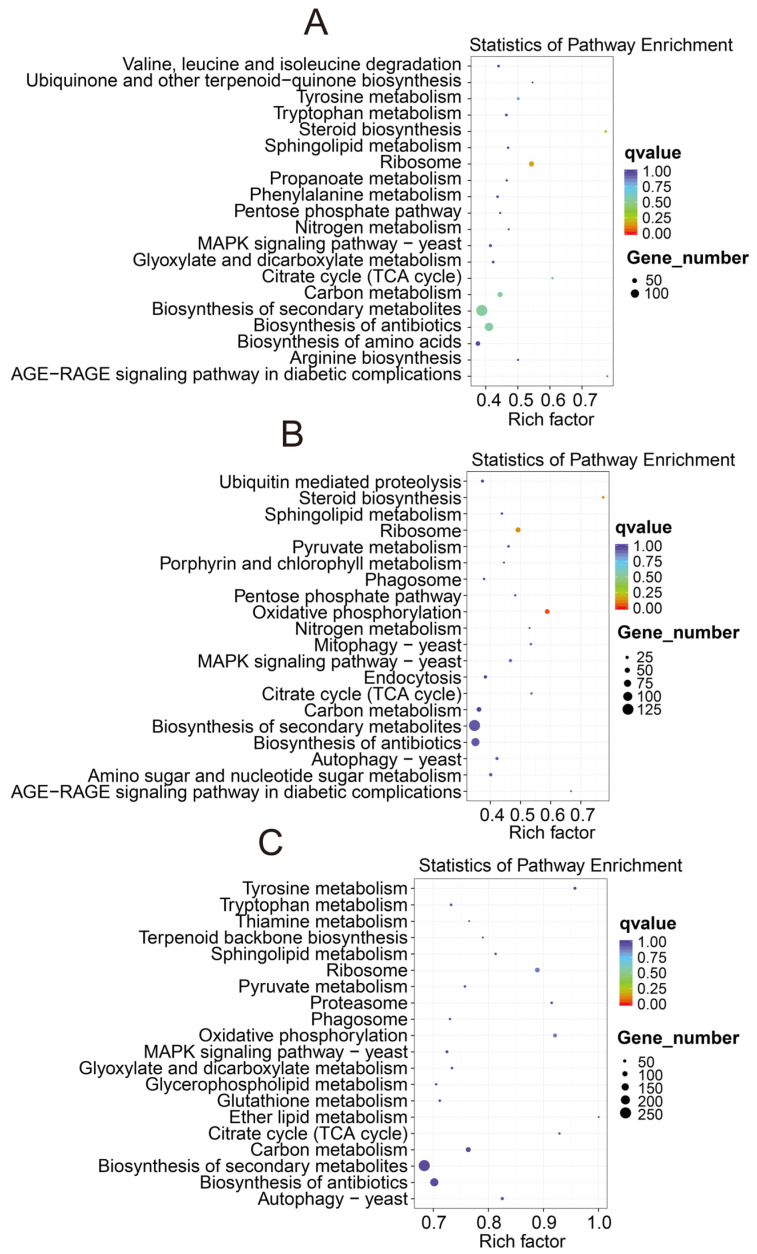
KEGG pathway analysis of differentially expressed genes in *A. phaeospermum* in three periods S2–S4. Note: Note: (**A**): S3 vs. S2; (**B**): S4 vs. S2; (**C**): S3 vs. S4. Note: The color of the dot represents different qvalues, and the size of the dot represents the number of differentially expressed genes in this pathway.

**Figure 9 biomolecules-12-01264-f009:**
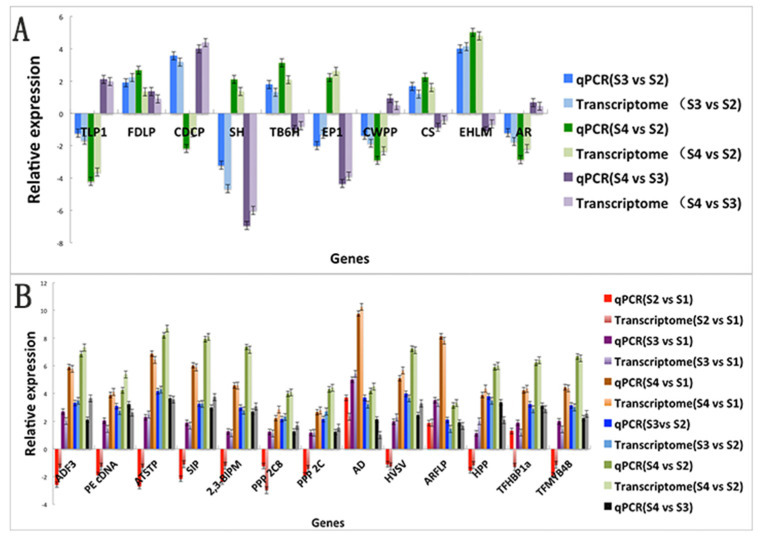
Fluorescence quantitative PCR results of differentially expressed genes in dual-seq. Note: (**A**) The 10 differentially expressed genes of *A. phaeospermum* under different infection periods were verified by fluorescence quantitative analysis; (**B**) The 13 differentially expressed genes of *B. pervariabilis* × *D. grandis* were verified by fluorescence quantitative analysis under different infection periods.

**Figure 10 biomolecules-12-01264-f010:**
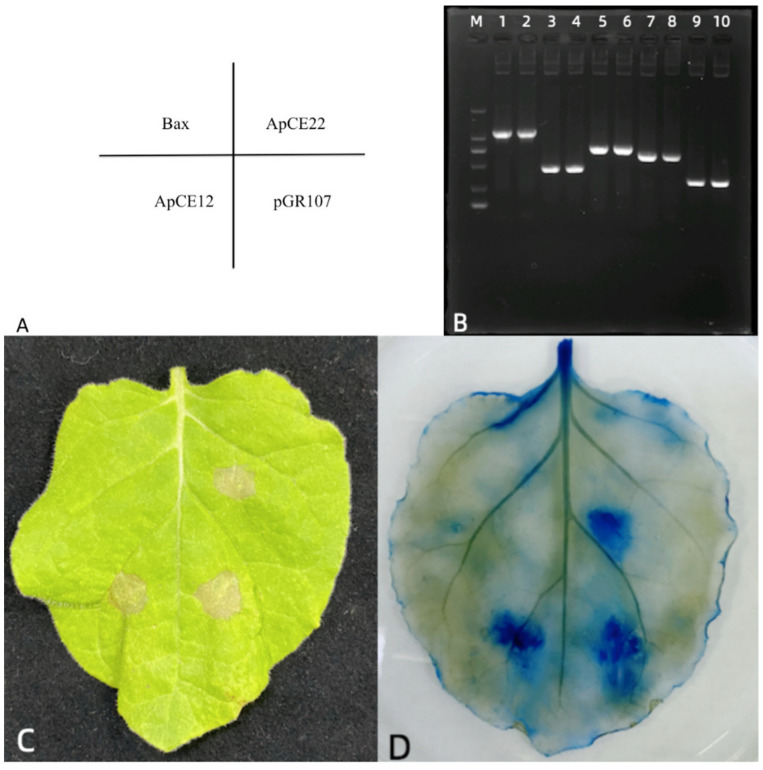
Transient expression results of effectors *ApCE12* and *ApCE22* in tobacco. Note: (**A**): Location map of tobacco leaf inoculation; (**B**): RT-PCR results of tobacco leaves inoculated with effectors; M: DL 2000 DNA marker; 1, 2: Tobacco Actin; 3, 4: *ApCE12*; 5, 6: *ApCE22*; 7, 8: Bax; 9, 10: pGR-107. (**C**): Effectors *ApCE12* and *ApCE 22* transient expression symptoms in tobacco. (**D**): Symptom of tobacco after trypan blue staining.

**Figure 11 biomolecules-12-01264-f011:**
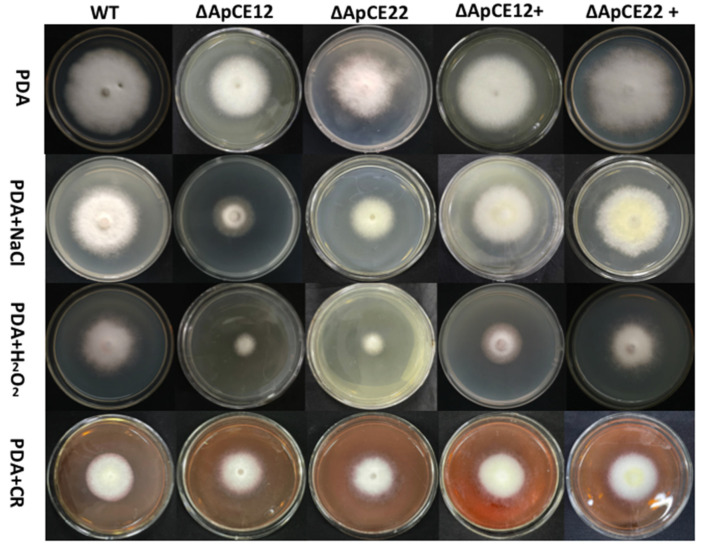
Comparison of colony morphology and stress tolerance of wild-type, Δ*ApCE12*, Δ*ApCE22*, *ApCE12* complementary strain, and *ApCE22* complementary strain. Note: The wild type, Δ*ApCE12*, Δ*ApCE22*, *ApCE12* complementary strain, and *ApCE22* complementary strain were inoculated on PDA medium or medium appended with various stress and cultured at 25 in darkness for 5 days.

**Figure 12 biomolecules-12-01264-f012:**
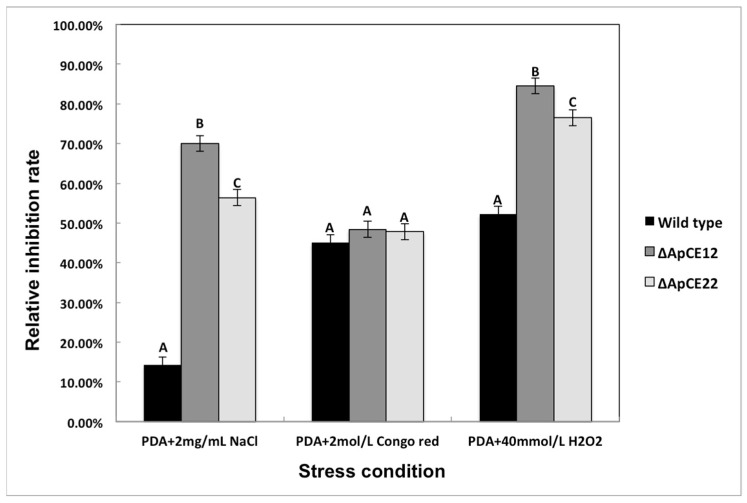
The bar chart shows the relative inhibition rate of wild-type, Δ*ApCE12*, and Δ*ApCE22* strains after NaCl and H_2_O_2_ stress for 5 days, respectively. Note: The datasets were calculated from the image in Figure 12. The error bars represent the standard deviations based on three independent biological replicates with three technical replicates each. The relative inhibition rates of Δ*ApCE12* and Δ*ApCE22* were significantly different from that of the wild type after NaCl and H_2_O_2_ stress. The relative inhibition rates of Δ*ApCE12* and Δ*ApCE22* were consistent with that of the wild type after CR stress. All assays were repeated three times; the data were analyzed using one-way ANOVA and Duncan’s range test in SPSS 16.0. Different capital letters (A, B and C) show that the relative inhibition rates of Congo red, NaCl, and H_2_O_2_ on the mycelial growth of different strains were significantly different. (*p* ≤ 0.01).

**Figure 13 biomolecules-12-01264-f013:**
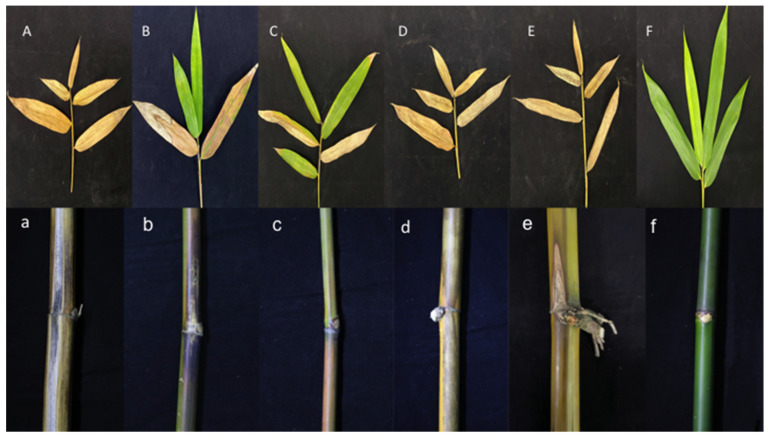
Symptoms of branches and leaves of plants infected with wild-type strain, Δ*ApCE12*, Δ*ApCE22*, *ApCE12* complementary strain, *ApCE22* complementary strain, and sterile water after 25 days. Note: (**A**,**a**) are the symptoms of *B. pervariabilis* × *D. grandis* leaves and branches inoculated with wild-type strain for 25 days, respectively. (**B**,**b**) are the symptoms of *B. pervariabilis* × *D. grandis* leaves and branches inoculated with Δ*ApCE12* strain for 25 days, respectively. (**C**,**c**) are the symptoms of *B. pervariabilis* × *D. grandis* leaves and branches inoculated with Δ*ApCE22* strain for 25 days, respectively. (**D**,**d**) are the symptoms of *B. pervariabilis* × *D. grandis* leaves and branches inoculated with *ApCE12* complementary strain for 25 days, respectively. (**E**,**e**) are the symptoms of *B. pervariabilis* × *D. grandis* leaves and branches inoculated with *ApCE22* complementary strain for 25 days, respectively. (**F**,**f**) are the symptoms of *B. pervariabilis* × *D. grandis* leaves and branches inoculated with sterile water for 25 days, respectively.

**Figure 14 biomolecules-12-01264-f014:**
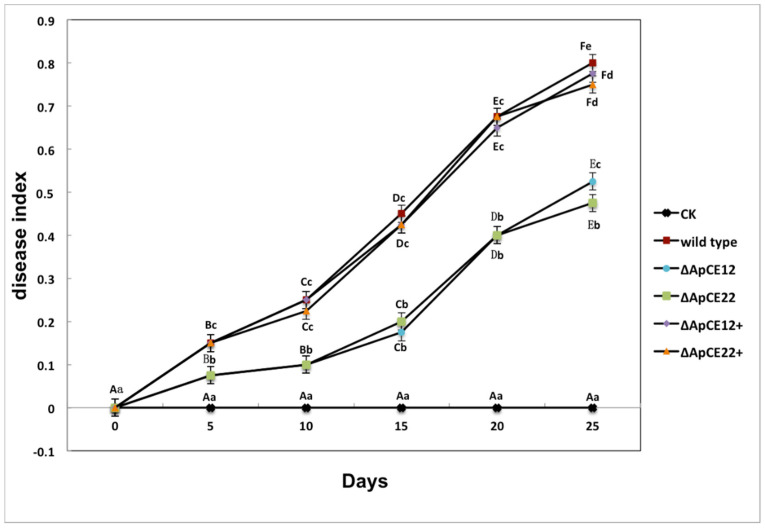
Dynamic changes in the disease index of *B. pervariabilis* × *D. grandis* infected by different strains. Note: CK, wild type, Δ*ApCE12*, Δ*ApCE22*, *ApCE12* complementary strain, and *ApCE22* complementary strain represent the changes in the disease index of *B. pervariabilis* × *D. grandis* inoculated with sterile deionized water, wild-type strain, *ApCE12* deletion mutant, *ApCE22* deletion mutant, *ApCE12* complementary strain, and *ApCE22* complementary *A. phaeospermum,* respective-ly. All assays were repeated three times; the data were analyzed using one-way ANOVA and Duncan’s range test in SPSS 16.0. Different lowercase letters indicate that the disease index of different strains in the same period is significantly different, and different capital letters indicate that the disease index of the same strain in different periods is significantly different (*p* ≤ 0.01).

**Table 1 biomolecules-12-01264-t001:** Sequencing quality test results of dual RNA-seq.

Sample	Raw Reads	Clean Reads	Effective Rate (%)	Q20 (%)	Q30 (%)	GC Content (%)
S1_1	54,580,508	51,965,959	95.21	98.19	94.55	53.41
S1_2	55,019,871	53,176,919	96.65	98.2	94.55	53.76
S1_3	53,461,164	51,508,083	96.35	98.12	94.28	53.49
S2_1	51,674,777	49,164,128	95.14	98.5	95.38	53.92
S2_2	54,082,748	52,079,582	96.3	98.52	95.38	53.94
S2_3	52,537,865	50,166,387	95.49	98.49	95.33	53.78
S3_1	52,623,574	50,656,315	96.26	98.56	95.53	54.41
S3_2	50,987,904	48,749,504	95.61	98.52	95.41	54.22
S3_3	52,046,192	49,929,476	95.93	98.59	95.58	54.16
S4_1	50,341,624	48,424,820	96.19	98.73	95.94	54.58
S4_2	55,324,267	52,522,315	94.94	98.46	95.21	54.52
S4_3	48,067,887	45,606,368	94.88	98.79	96.08	54.48

**Table 2 biomolecules-12-01264-t002:** The results of clean reads alignment *A. phaeospermum* genome in dual RNA-seq sequencing.

Sample	Total Reads	Total Mapped	Multiple Mapped	Uniquely Mapped
S1_1	148,594,824	4726 (0%)	2591 (0%)	2135 (0%)
S1_2	136,746,070	2489 (0%)	1559 (0%)	930 (0%)
S1_3	146,740,098	3722 (0%)	2082 (0%)	1640 (0%)
S2_1	126,643,902	7,306,133 (5.77%)	23,757 (0.02%)	7,282,376 (5.75%)
S2_2	137,144,488	7,537,105 (5.5%)	28,209 (0.02%)	7,508,896 (5.48%)
S2_3	132,487,818	6,700,653 (5.06%)	23,231 (0.02%)	6,677,422 (5.04%)
S3_1	169,973,228	38,580,876 (22.7%)	136,813 (0.08%)	38,444,063 (22.62%)
S3_2	126,157,938	29,676,163 (23.52%)	118,045 (0.09%)	29,558,118 (23.43%)
S3_3	132,852,916	32,475,890 (24.44%)	124,742 (0.09%)	32,351,148 (24.35%)
S4_1	120,447,454	60,467,620 (50.2%)	139,444 (0.12%)	60,328,176 (50.09%)
S4_2	106,076,392	51,212,956 (48.28%)	118,143 (0.11%)	51,094,813 (48.17%)
S4_3	167,634,532	82,155,824 (49.01%)	169,114 (0.1%)	81,986,710 (48.91%)

**Table 3 biomolecules-12-01264-t003:** Results of clean reads alignment *B. pervariabilis* × *D. grandis* in dual RNA-seq sequencing.

Sample	Total Reads	Total Mapped
S1_1	147,723,068	147,723,068 (74.11%)
S1_2	136,170,664	136,170,664 (74.35%)
S1_3	145,945,350	145,945,350 (74.99%)
S2_1	126,064,378	126,064,378 (71.50%)
S2_2	136,324,086	136,324,086 (72.18%)
S2_3	131,764,534	131,764,534 (72.98%)
S3_1	168,734,396	168,734,396 (58.43%)
S3_2	125,003,468	125,003,468 (58.29%)
S3_3	131,647,634	131,647,634 (57.61%)
S4_1	119,832,860	119,832,860 (36.16%)
S4_2	105,269,626	105,269,626 (37.57%)
S4_3	166,986,336	166,986,336 (36.77%)

## Data Availability

All interaction transcriptome data of 12 samples of *B. pervariabilis* × *grandis* were deposited in the NCBI Sequence Reads Archive (SRA) under the accession numbers SRR14685222, SRR14685221, SRR14685220, SRR14685219, SRR14685218, SRR14685217, SRR14685216, SRR14685215, SRR14685214, SRR14685213, SRR14685212, and SRR14685211. The raw data were published in the NCBI Sequence Read Archive (SRA) under the Bio Project acces-sion number SAMN19312317 (https://www.ncbi.nlm.nih.gov/biosample/SAMN19312317/ accessed on 1 September 2022).

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
