# Peer review of "Functional Identification of Arthrinium phaeospermum Effectors Related to Bambusa pervariabilis × Dendrocalamopsis grandis Shoot Blight"

_biomolecules, 2022, doi:10.3390/biom12091264_

Round 1

Reviewer 1 Report

Comment 1: Change the title

Comment 2: Abstract should be understandable for new readers, include an introduction, materials and methods, results and discussion, and conclusion. Please rewrite the abstract

Comment 3: Line no. 36-37, rewrite the sentence

Comment 4: While writing scientific name, only first-time mention genus name, second time no need to mention, for example, check line no. 68-Ustilago maydis and line no. 83, Aspergillus flavus, and Zea mays should be in italics, check throughout the manuscript.

Comment 5: Title about Dual seq analysis but abstract started like that bu there is no basic information or previous research about Dual seq analysis in the introduction. Please include some information.

Comment 6: Please give details abbreviations, for example GO, FDR, RPKM, KEGG and so on. Please check throughout the manuscript.

Comment 7: Remove the materials at line no. 130 and methods at line no. 13d, check author's instructions

Comment 8: Lot of information, new studies, and so many things in the results section but the discussion is not up to the level. Please improve the discussion section.

Comment 9: If possible please provide separate conclusion section.

Reviewer 2 Report

The manuscript describes the function of Arthrinium phaeospermum effectors screened by RNA-seq analysis. First, RNA-seq analysis of hybrid bamboo infected with Arthrinium phaeospermum is performed to analyze the pathogen and plant transcriptomes. Next, the screened Arthrinium phaeospermum effectors are expressed in tobacco, and observation of programmed cell death is performed. The manuscript also generates Arthrinium phaeospermum mutants lacking effector genes and analyzes stress tolerance and pathogenicity of the mutants.

This manuscript contains novel findings on the interaction between Arthrinium phaeospermum and hybrid bamboo. However, the following issues should be considered and revised prior to the acceptance of the manuscript.

1.       Cell death in effector-expressing tobacco has not been clearly demonstrated (fig. 10). Cell death or DNA fragmentation should be observed by trypan blue staining or the TUNEL method.

2.       The manuscript contains many figures and tables. Please consider moving some of them to the supplementary materials. For example, Tables 1-4 showing primer sequences and Figure 11 of protoplast observations.

3.       The manuscript has some mistakes, so please check the entire document carefully.

Line 22 symtoms.nMutant -> symtoms. Mutant

Line 128 interaction -> interaction.

Line 608 107 -> Is 7 superscript?

Round 2

Reviewer 2 Report

The revised manuscript has addressed the issues raised in the previous review. The manuscript is now acceptable for publication.